# ATOMS (Adjustable Transobturator Male System) Is an Effective and Safe Second-Line Treatment Option for Recurrent Urinary Incontinence after Implantation of an AdVance/AdVance XP Fixed Male Sling? A Multicenter Cohort Analysis

**DOI:** 10.3390/jcm11010081

**Published:** 2021-12-24

**Authors:** Fabian Queissert, Keith Rourke, Sandra Schönburg, Alessandro Giammò, Andreas Gonsior, Carmen González-Enguita, Antonio Romero, Andres J. Schrader, Francisco Cruz, Francisco E. Martins, Juan F. Dorado, Javier C. Angulo

**Affiliations:** 1Department of Urology and Pediatric Urology, University of Münster, Schlosspl. 2, 48149 Münster, Germany; Fabian.Queissert@ukmuenster.de (F.Q.); AndresJan.Schrader@ukmuenster.de (A.J.S.); 2Department of Urology, Alberta University, 116 St. & 85 Ave., Edmonton, AB T6G 2R3, Canada; krourke@ualberta.ca; 3Department of Urology and Kidney Transplantation, Martin Luther University, Universitätsplatz 10, 06108 Halle (Saale), Germany; sandra.schoenburg@uk-halle.de; 4Department of Neuro-Urology, CTO/Spinal Cord Unit, AOU Città della Salute e della Scienza di Torino, Via Gianfranco Zuretti, 29, 10126 Torino, Italy; giammo.alessandro@gmail.com; 5Klinik und Poliklinik für Urologie, University of Leipzig, Augustuspl. 10, 04109 Leipzig, Germany; Andreas.Gonsior@medizin.uni-leipzig.de; 6Department of Urology, Hospital Universitario Fundación Jiménez Díaz, Av. de los Reyes Católicos, 2, 28040 Madrid, Spain; CGEnguita@fjd.es; 7Department of Urology, Hospital Universitario Morales Meseguer, Av. Marqués de los Vélez, s/n, 30008 Murcia, Spain; antonio.romero6@carm.es; 8Centro Hospitalar São João, Faculty of Medicine of Porto and i3S Institute, Alameda Prof. Hernâni Monteiro, 4200-319 Porto, Portugal; cruzfjmr@med.up.pt; 9Centro Hospitalar Universitário de Lisboa Norte, Hospital Santa María, Av. Prof. Egas Moniz MB, 1649-028 Lisboa, Portugal; pascual-uro@hotmail.com; 10PeRTICA Statistical Solutions, Av. Leonardo Da Vinci, 8, OF217, Getafe, 28906 Madrid, Spain; jfdorado@pertica.es; 11Clinical Department, Faculty of Biomedical Science, Universidad Europea de Madrid, Carretera de Toledo, Km 12.500, Getafe, 28905 Madrid, Spain

**Keywords:** adjustable transobturator male system, stress urinary incontinence, artificial urinary sphincter, fixed male sling, sling failure

## Abstract

(1) Background: This study examined outcomes of second-line ATOMS implantation after failure of the fixed male sling (FMS) AdVance/AdVance XP. (2) Methods: A retrospective multicenter cohort analysis was carried out in men implanted with an ATOMS between 2011 and 2020 after failure of an AdVance/AdVance XP. Success was assessed on the basis of objective (dryness, 0–1 pad/24 h or >20 g/24 h pad test) and subjective results (PGI-I). We performed the Wilcoxon rank sum test, Fisher’s exact test, logistic regression, and multivariate analysis. (3) Results: The study included 88 patients from 9 centers with a mean age of 71.3 years. No Clavien–Dindo > II complications occurred within the first 3 months after ATOMS implantation. A total of 10 cases (9%) required revision in the ensuing clinical course. After a mean follow-up of 42.5 months, 76.1% achieved social continence, and 56.8% used no pads at all. Mean urine leakage/24 h dropped from 422 g (3.9 pads) to 38 g (0.69 pads) and the mean ICIQ-SF decreased from 16.25 to 5.3 (*p* < 0.0001). PROMs (patient-reported outcome measures) showed improvement in 98.9% of cases, and 63.6% gave a “very much better” PGI-I rating. Multivariate analysis identified a lower probability of achieving maximum satisfaction for the following factors: the AdVance XP as first-line therapy (OR 0.35), a lower ICIQ-SF question 1 (OR 0.26), status post-irradiation (OR 0.14), and more severe pain prior to ATOMS implantation (OR 0.51). (4) Conclusions: Implantation of an ATOMS is an effective and safe second-line treatment option for recurrent urinary incontinence after implantation of an AdVance/AdVance XP sling. High patient satisfaction was demonstrated in a long-term follow-up.

## 1. Introduction

Various surgical concepts are available for treating stress urinary incontinence (SUI) after radical prostatectomy. The guidelines of the International Continence Society (ICS) still recommend as preferred treatment the artificial urinary sphincter (AUS). Fixed male slings (fMS) are mentioned as alternatives for patients with only mild to moderate SUI [1]. The AdVance/AdVance XP (Boston Scientific, Marlborough, MA, USA) are the most commonly implanted fMS and are used in cases with a mobile bulbar urethra and intact urethral sphincter with coaptation zone >1 cm. If properly applied and correctly indicated, its retrourethral placement leads to dorsal stabilization of bulbar urethra and bulging and repositioning of the membranous urethra with consecutive improvement of functional length [2]. Cure rates for fMS range between 8.3% and 87%, depending on the definition of the success parameter “cured” and the severity of preexisting incontinence [3,4]. The MASTER trial comparing AUS and fMS based the assessment of treatment success on patient-reported outcome measures (PROMs) in addition to objective criteria and found that 72.2% of patients in the fMS arm were completely or fairly satisfied after a follow-up of 1 year [5]. Since the effect of an fMS is known to weaken years after the intervention, the percentage of satisfied patients could decrease even further in clinical course [4].

Analysis of repeat fMS implantation yielded a failure rate of 55% [6]. Rheder et al. likewise demonstrated a poorer outcome for second-line application of the AdVance [7], and Soljanik et al. reported a tolerable outcome after a short follow-up of 3 months [8]. Munier et al. reported a 66.7% continence rate for second-line implantation of an adjustable ProACT [9], but Baron et al. documented a 28% revision rate in that scenario [10]. Descriptive studies on second-line use of the AUS reported an acceptable outcome [6,11,12], but also with a relatively high revision rate [13]. It must be taken into account that, if given the choice, patients tend to choose the fMS rather than the AUS because of the easy handling and the less complex intervention [14].

The adjustable transobturator male system (ATOMS, A.M.I. GmbH, Feldkirch, Austria) provides a further option for both the first-line treatment of SUI and second-line treatment for recurrent incontinence after failure of male slings. It is placed under the bulbar urethra and thus is somewhat more distal than the fMS. The silicone cushion compresses the corpus spongiosum and likewise leads to bulging of the membranous urethra [15]. However, it differs from the fMS in several respects. Unlike the fixed sling, it can even be used in patients with a partially damaged sphincter, which enlarges its application range. The mode of action is a mixture of direct compression of the bulbar urethra and, in case of good urethral elasticity, a distribution of pressure to the membranous urethra [16]. The scrotal port allows simple adjustments, which can even be done years later. Broadly applied, it has already achieved better continence rates than fMS [4] and lower complication rates than the AUS [17,18]. However, use of the ATOMS after failed fMS implantation has thus far only been investigated in small cohorts [18,19]. Our study analyzes, for the first time, a large multicenter cohort of patients who received an ATOMS after failed AdVance/AdVance XP implantation. We identify risk factors for poorer outcome and propose a nomogram to facilitate the indication of ATOMS as a second-line treatment.

## 2. Materials and Methods

### 2.1. Study Population

This is a retrospective cohort multicenter study evaluating the effectiveness and safety of ATOMS after failed fMS Advance/Advance XP. Inclusion criteria were persistent bothersome SUI for more than 6 months after previous mFS and ATOMS implantation as rescue surgery with a minimum 6 month follow-up. Use of other incontinence devices before fMS and predominant detrusor overactivity were exclusion criteria. Radiotherapy or previous history of bladder neck stricture or bulbar urethroplasty for urethral stenosis were not exclusion criteria provided that cystoscopy ruled out current obstruction. SUI severity and patient age were not limiting factors for inclusion either. The study was derived from current clinical practice. The indication for ATOMS and not another device was made by physician with the informed consent of the patient. IRB approval was obtained (2020-823-f-S).

The surgical technique for ATOMS placement followed the original description of Seweryn et al. [15]. The fMS was divided at its mid-portion to palpate the urethra in midline with the Foley catheter and with special attention not to damage the urethra. Careful hemostasis was always performed and drainage was not placed. When necessary, postoperative adjustment was performed in the office 2–3 weeks after the implantation by percutaneous injection of physiological sodium chloride solution through the port membrane and thereafter when required at intervals of 4 weeks until either dryness was achieved or maximum filling capacity of the system was reached.

### 2.2. Study Endpoints

The primary endpoint was the evaluation of effectiveness and safety of ATOMS as a secondary implant to treat male SUI after failed fMS Advance/Advance XP. Multivariate analysis was performed to determine factors that determine a “very much better” PGI-I rating (highest satisfaction with the ATOMS implant).

### 2.3. Variables Evaluated

Data analyzed included demographics, previous radiotherapy, former history of bladder neck stenosis and urethral stricture, intraoperative and postoperative complications, continence outcomes and follow-up. SUI severity of incontinence (pad count, pad test, ICIQ-SF) was registered at baseline before the Advance/Advance XP, after the Advance/Advance XP surgery and after ATOMS adjustment. Visual analogue scale (VAS) for pain on a 0–10 scale was registered both after Advance/Advance XP and after ATOMS surgery at the day of hospital discharge, and scores were compared.

Postoperative complications were defined following the Clavien–Dindo classification within 3 months after surgery. Continence outcomes were evaluated at the time when adjustment was considered complete. Patients who used none or one safety pad/day with a 24-h pad-test ≤ 20 mL/day were considered dry. Patients with zero pads/day were also analyzed. We defined an improvement in our study when the self-assessed patient global impression of improvement (PGI-I) was specified as 1, “very much better than before” or 2, “much better than before”. No patient was lost to follow-up.

### 2.4. Statistical Analysis

Statistics were calculated as the median values, interquartile range (IQR), minimum and maximum for continuous variables, and as the frequency and percent for categorical data. Differences were calculated by the Wilcoxon rank sum test for continuous variables and the Fisher exact test for categorical variables. A *p* value < 0.05 was considered significant. Logistic regression was performed using a stepwise model (entry 0.15 and stay criterium 0.1) to evaluate independent variable determinants of dryness (no pad or one safety pad/day with a 24-h pad-test ≤ 20 mL/day). Multivariate analysis was also performed to detect the variables related to the previous Advance/Advance XP implant that independently predict the best patient perception based on PGI-I with rescue ATOMS, and a nomogram is proposed to predict the probability to achieve this result. The statistical analysis was developed using Statistical Analysis System 9.3 (SAS Institute Inc., Cary, NY, USA).

## 3. Results

A total of 88 patients intervened with ATOMS between 2011 and 2020 in 9 academic institutions were included in the study. Table 1 summarizes clinical data. The mean age at time of ATOMS placement was 71.3 ± 5.9 (range 53–85) years and mean follow-up was 42.5 ± 20 (range 7–108) months. Besides prostatectomy, 18 patients (20.5%) also received radiotherapy and 14 (15.9%) had previous history of treated bladder neck stricture (5 cases) or bulbo-membranous urethral stenosis (9 cases). The Advance (47 cases; 53.4%) or Advance XP (41 cases; 46.6%) had been previously implanted at a mean of 40.3 ± 34.1 (range 2–216) months before ATOMS.

### 3.1. Effectiveness of Secondary ATOMS Implant

A total of 67 patients (76.1%) used no pad or one security pad with ≤20 mL pad test/24-h urine loss and were considered “dry”. Inasmuch, 50 patients (56.8%) used no pads after ATOMS. Adjustment was completed after a mean 2.8 ± 2.1 postoperative fillings, reaching a mean 15.4 ± 5.4 mL filling volume. The failed Advance/Advance XP implantation showed a deterioration both in pad-test and ICIQ-SF compared to the baseline situation (*p* < 0.001 and *p* < 0.0001, respectively), while ATOMS implantation caused a positive effect in both outcomes (*p* < 0.0001 each) (Figure 1 and Figure 2).

A very significant improvement was observed both in SIU severity and PROMs after ATOMS adjustment. ATOMS implantation changed mean 24-h pad test from 422 ± 280 (range 30–2000) mL to 38 ± 97 (range 0–580) mL; mean Δ 384 ± 231 (range 0–1450) mL (*p* < 0.0001). Similarly, ICIQ-SF changed from 16.25 ± 2.5 (range 10–21) to 5.3 ± 3.3 (range 1–15); mean Δ 11 ± 3.4 (range 1–17) (*p* < 0.0001). Equivalent improvements were evidenced by each ICIQ-SF question evaluated. Mean pad count changed from 3.9 ± 1.5 (range 2–8) pads/24-h to 0.69 ± 0.9 (range 0–4) (*p* < 0.0001) and night pad count changed from 0.7 ± 0.8 (range 0–3) to 0.1± 0.25 (range 0–1) pads (*p* < 0.0001) (Table 1).

Multivariate analysis for factors that predict dryness revealed that the absence of previous radiotherapy (OR 6.6 (95% CI 1.8–24.4); *p* = 0.0004), ICIQ-SF ≤ 16 (OR 5.6 (95% CI 1.5–20.2); *p* = 0.003) and patient age ≥70 years (OR 3.8 (95% CI 1.1–13.05); *p* = 0.026) predicted the best objective results, which were defined as a pad test ≤20 mL. Other variables investigated, such as type of Advance (Advance/Advance XP), pad test, pad count, night pad count, number of fillings of ATOMS, total filling of the ATOMS system and need of surgical revision during follow-up were not predictors of dryness. The area under the curve (AUC) for this predictive model was 0.83.

### 3.2. Safety of Secondary ATOMS Implant

There were no intraoperative complications at the time of ATOMS implant after failed Advance. A total of 7 patients (7.95%) had early postoperative complications: hematoma (*n* = 3), urinary infection, severe pain, hyperesthesia and acute urinary retention (1 each, respectively). All were minor according to Clavien–Dindo classification (4 cases grade I and 3 grade II) and none implied surgical revision within the first 3 months. During follow-up, 10 cases (9%) were explanted because of device infection (4 cases), persistent incontinence (4 cases), port erosion (1 case) and persistent pain (1 case).

Regarding pain at discharge, 62 patients (70.45%) classified pain as “0” after ATOMS surgery. The same score had been given by 41 (46.6%) after Advance/Advance XP implantation. Mean VAS for pain value registered was 0.9 ± 1.3 (range 0–8) after fMS and 0.7 ± 1.4 (range 0–6) after ATOMS, thus revealing significantly less intensity of pain at discharge for ATOMS (*p* < 0.0001) (Table 1, Figure 3).

### 3.3. Patient Reported Outcome Measures (PROMs) of Secondary ATOMS Implant

A total of 87 cases (98.9%) reported improvement after ATOMS, and only one case declared “same as before”. 56 patients (63.6%) self-declared “very much better than before”, 23 (26.1%) “much better than before” and 8 (9.1%) “better than before”. Mean PGI-I score was 1.5 ± 0.7 (range 1–4). 

Multivariate analysis for factors related to the previous Advance/Advance XP implant that predict best patient perception (PGI-I = 1) after ATOMS revealed that absence of radiotherapy (OR 0.14 (95% CI 0.04–0.52); *p* = 0.003), lower baseline ICIQ-SF Question 1 response regarding frequency of urine loss (as per score) (OR 0.26 (95% CI 0.1–0.67); *p* = 0.005), lower VAS of pain before ATOMS (as per score) (OR 0.51 (95% CI 0.31–0.86); *p* = 0.01) predict best patient perception with rescue ATOMS. The type of previous fMS (Advance compared to Advance XP) (OR 0.35 (95% CI 0.12–1.04); *p* = 0.06) is another variable with influence in best patient perception after ATOMS but does not reach statistical significance. The AUC for this predictive model was 0.81. Based on these results, a nomogram to predict best patient perception with ATOMS after a failed Advance system is proposed (Figure 4).

## 4. Discussion

Our study demonstrates for the first time that ATOMS is highly effective even after failed fMS implantation. Full continence was achieved in 56.8% of patients, and a total of 76.1% used no or one pad a day. Reduction of urine leakage from a mean of 422 to 38 g or from 3.9 to 0.69 pads in the 24 h pad test was also reflected in the patients’ subjective evaluation of treatment success. The PGI-I rating was “very much better” or “much better” in 89.7% of the cases. Application of ATOMS after failed fMS implantation thus shows effectiveness comparable to that found in a current review with a dry rate of 67% (0–1 pad/24 h) and an improvement rate of 90% (>50%) [18]. This also confirms results from a small single-center study on ATOMS implantation after failed fMS by Angulo et al., who found a dry rate of 81.8% (0–1 pad/24 h) [19]. In an earlier study, Friedl et al. documented a higher surgical revision rate for the ATOMS, but they included mostly the 1st generation ATOMS Inguinal Port [20]. The current ATOMS device has a pre-attached and silicone-covered inguinal port that facilitates postoperative filling for a safer and easier device adjustment [21].

Comparison with other implants in second-line application is not always easy due to the often fundamentally different initial situations. Several authors investigated fMS as a second-line option and mostly obtained poorer results than after its primary application. In particular, high expertise of the surgeons who implanted the first fMS [6] and a short interval between implantation and revision [8] exert an additional negative influence on outcome. Moreover, publications could have overestimated the success of the fMS as a second-line option because of the very short follow-up. As the gold standard of surgical treatment for SUI to date, the AUS has surely been used as the second-line option in most cases worldwide, and various study groups have demonstrated a favorable outcome. Ajay et al. found a dry rate of 94% for the AUS (0–1 pad, <8 g/24 h) and of only 45% for repeated fMS after a short follow-up of 4.5 months [6]. Lentz et al. documented a continence rate of 96% after 3 months [11]. Ziegelmann et al. evaluated the impact of prior urethral sling on AUS outcomes and recognized that the revision rate of AUS tended to be higher for second than for primary implantation (70% vs. 85% revision-free survival after 3 years) [22]. Due to circular compression, the AUS carries the highest risk of urethral erosion of all implants, particularly in view of the real-life perspective with centers of varying expertise [23,24] and taking into account a long follow-up [13,22].

In contrast, urethral erosion is rare with the ATOMS [18] and has thus far only been described in two cases [19]. In our analysis, there were a few minor complications within the first three months. Placement of the ATOMS is comparatively unproblematic due to the somewhat more distal positioning of its cushion under the bulbus. Additionally, differing from the fMS in its peribulbar placement, the ATOMS only requires a small incision for the passage of the helical needles used to place the mesh arms. For this purpose, uncomplicated severance of the old fMS was carried out in all cases. Revision was necessary in 9% of the patients during follow-up of 42.5 months, with the most common reasons being infection of the implant and persistent incontinence.

Chronic pain in the implant area is considered a complication common to both fMS and adjustable slings. Meisterhofer et al. described a rate of 1.3% for fixed slings and 1.5% for adjustable systems [4]. We found a mean VAS of 0.7 after ATOMS as a secondary implantation. Only one revision of the ATOMS was performed because of persistent pain. After primary implantation of fMS, on the other hand, the mean VAS was 0.9 in our cohort, which signifies a significantly higher level of pain at hospital discharge. However, a limitation of our retrospective work is the lack of pain evaluation data during follow-up. Despite that, the role pain plays became apparent in our multivariate analysis of patient satisfaction. Pain prior to ATOMS implantation reduced the probability of maximum satisfaction (PGI-I = 1) by nearly 50%. The lowest probability of a PGI-I = 1 rating was found for irradiated patients (OR 0.14). Though detecting this connection only confirms the results of all other studies on ATOMS [17,18,20], we were also able to identify ICIQ-SF question 1 (frequency of urine leakage) as a further strong predictor of maximum patient satisfaction (OR 0.26). This is in line with Machioka et al. [25], and we likewise regard this well-designed questionnaire as a valuable aid in assessing incontinence. In the multivariate analysis, revision after implantation of an AdVance rather than an AdVance XP was also associated with a higher probability of maximum satisfaction (OR 0.35). This could be due to a higher percentage of patients with a loosened sling after implantation of an AdVance, since technical modifications are meant to confer higher stability to the AdVance XP [26]. The more mobile bulbar urethra resulting from the loosening of the AdVance enables better positioning of the ATOMS and could enhance its functionality through increased urethral elasticity. 

In the multivariate analysis of factors influencing the objective continence situation, irradiation (OR 6.6) and a high total ICIQ-SF (OR 5.6) were again associated with a lower probability to achieve continence. In contrast to the analysis published by Husch et al. [23], lower age was also associated with a poorer objective outcome (OR 3.8). Lifestyle may play a role here. The active lifestyle regained by continence could lead to somewhat higher urine leakage in this group of patients, while at the same time explaining their high satisfaction with the ATOMS in spite of this. Meisterhofer et al. also reported a difference between subjective and objective cure rates when using fMS [4]. The idea that assessment of treatment success should not be based solely on the absolute numbers from a 24 h pad test has already been discussed according to the results of the MASTER trial [5]. We also attached great importance to patient satisfaction and propose a nomogram including predictive variables. By combining them, the probability of maximum satisfaction with ATOMS as a second-line option after failed fMS can be predicted.

Studies dealing with surgical interventions can be subject to multiple biases. However, the surgical technique of ATOMS implantation is simple, and therefore, great deviations among surgeons cannot be assumed. Additionally, all the surgeons involved in the study have wide experience with the implantation of the device. Besides, the results presented are based on real-world data and, in this sense, the variabilities derived from different surgeons and institution are not in fact a limitation of the investigation. Probably, there is a higher variability in the appropriateness of the surgical technique for previous AdVance implantation, and also in the reasons for AdVance failure. However, the conclusion derived from this study is that ATOMS is an effective and safe option after previously failed AdVance. The main limitation of this study stands in its retrospective nature. On the other hand, the long follow-up, the multicenter cohort and the concomitant analysis of objective parameters and PROMs are strengths of our analysis. Furthermore, although it needs validation, the nomogram we present may serve to guide decisions regarding whether ATOMS is a good option in a particular patient. Prospective randomized studies that compare the different options after failed AdVance system are needed. 

## 5. Conclusions

ATOMS strongly reduces urine leakage even in revision surgery after failure of a fMS, and globally, 89.7% of men with an ATOMS after failed AdVance/AdVance XP considered themselves “very much better” or “much better”. Despite the relatively long mean follow-up of the series, there were no cases of urethral erosion, and the revision rate was tolerable. Apart from the known risk factors of previous irradiation and the ICIQ-SF to modulate the results of an ATOMS, we were able, for the first time, to demonstrate the association with preexistent chronic pain after Advance/Advance XP implantation as well as with the type of implant previously used. Based on these findings, we propose a nomogram that can help predict satisfaction with ATOMS as a secondary implant and could be useful as a decision-making tool for determining whether patients with a failed fMS could benefit from an ATOMS or would require an AUS.

## Figures and Tables

**Figure 1 jcm-11-00081-f001:**
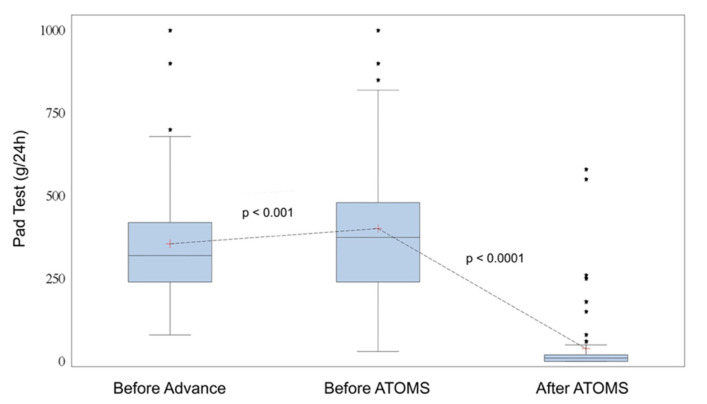
Pad test evolution before Advance, before ATOMS and after ATOMS.

**Figure 2 jcm-11-00081-f002:**
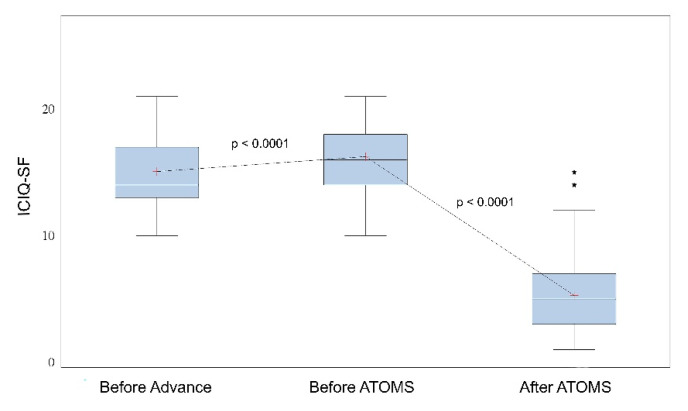
ICIQ-SF total score evolution before Advance, before ATOMS and after ATOMS.

**Figure 3 jcm-11-00081-f003:**
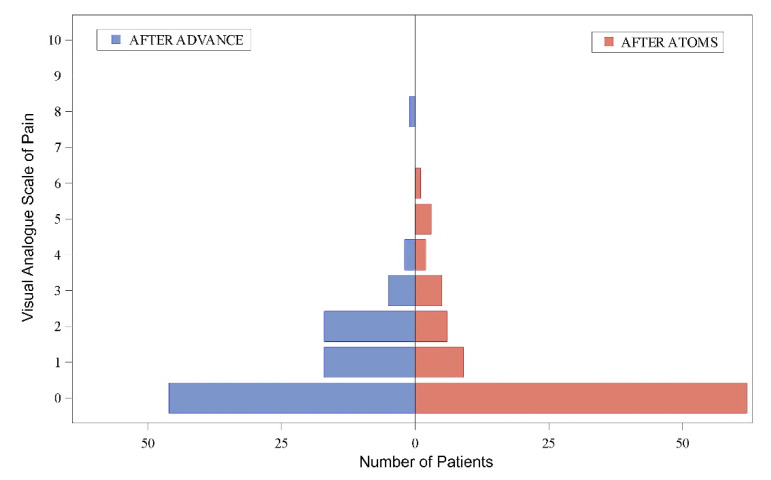
Visual analogue scale for pain at hospital discharge after Advance (**left**) and ATOMS implantation (**right**).

**Figure 4 jcm-11-00081-f004:**
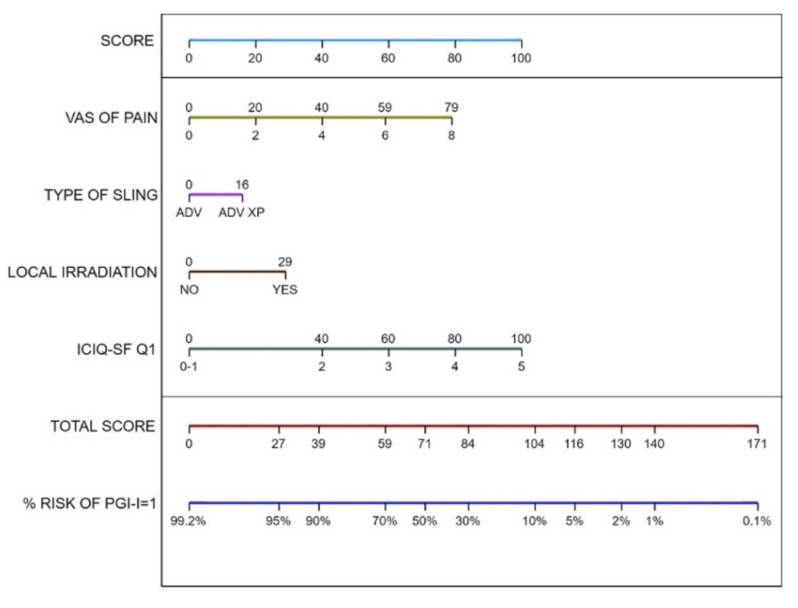
Nomogram to predict best patient rating (PGI-I = 1, “very much better”) with rescue ATOMS in patients with persistent incontinence after Advance.

**Table 1 jcm-11-00081-t001:** Data of patients undergoing ATOMS after the Advance system (*n* = 88).

Variable	*n* (%)
Data before Advance implant	
Type of retrourethral sling, *n* (%)	
Advance	18 (23)
Advance XP	24 (30.8)
24-h pad test, mL, mean ± SD (range) (*)	355 ± 194 (80–1200)
24-h pad count, *n*, mean ± SD (range) (*)	3.4 ± 1.3 (1–6)
Night pad count, *n*, mean ± SD (range) (*)	0.4 ± 0.7 (0–2)
ICIQ-SF total (*)	15.1 ± 3.8 (10–32)
ICIQ-SF Question 1 (*)	3.4 ± 0.8 (2–5)
ICIQ-SF Question 2 (*)	4.5 ± 2.3 (2–18)
ICIQ-SF Question 3 (*)	7.1 ± 1.9 (4–10)
VAS for pain (0–10), mean ± SD (range) (*, #)	0.9 ± 1.3 (0–8)
Data before ATOMS implant	
Age, years, mean ± SD (range)	71.3 ± 5.9 (53–85)
Previous radiation, *n* (%)	18 (20.5)
Previous urethral/bladder neck strictures, *n* (%)	14 (15.9)
Time since sling surgery, months, mean ± SD (range)	40.3 ± 34.1 (2–216)
24-h pad test, mL, mean ± SD (range)	422 + 280 (30–2000)
24-h pad count, *n*, mean ± SD (range)	3.9 ± 1.5 (2–8)
Night pad-count, *n*, mean ± SD (range)	0.7 ± 0.8 (0–3)
ICIQ-SF total	16.25 ± 2.5 (10–21)
ICIQ-SF Question 1	3.9 ± 0.7 (2–5)
ICIQ-SF Question 2	4.7 ± 1.1 (2–6)
ICIQ-SF Question 3	7.6 ± 1.5 (4–10)
Perioperative complication, *n* (%)	0 (0)
Postoperative complications, *n* (%)	7 (7.95)
VAS for pain (0–10), mean ± SD (range) (#)	0.7 ± 1.4 (0–6)
Total filling volume, mL, mean ± SD (range)	15.4 ± 5.4 (6–27)
Number of fillings, *n*, mean ± SD (range)	2.8 ± 2.1 (0–8)
Data after ATOMS adjustment	
Follow-up after ATOMS, months, mean ± SD (range)	42.5 ± 20 (7–108)
Patients with pad test < 20 mL, *n* (%)	67 (76.1)
Patients with zero pad test, *n* (%)	50 (56.8)
24-h pad test, mL, mean ± SD (range)	38 ± 97 (0–580)
24-h pad count, *n*, mean ± SD (range)	0.6 ± 0.9 (0–4)
Night pad count, *n*, mean ± SD (range)	0.1 ± 0.25 (0–1)
ICIQ-SF total	5.3 ± 3.3 (1–15)
ICIQ-SF Question 1	1.4 ± 1.2 (0–4)
ICIQ-SF Question 2	1.2 ± 1.2 (0–4)
ICIQ-SF Question 3	2.65 ± 1.5 (1–7)
PGI-I scale, *n* (%)	
Very much better	56 (63.6)
Much better	23 (26.2)
Better	8 (9.1)
Same as before	1 (1.1)

ATOMS, adjustable transobturator male system; SD, standard deviation. (*) Data evaluated in a population of 79 patients. (#) VAS visual analogue scale at hospital discharge.

## Data Availability

Full data will be available upon reasonable request to corresponding author.

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
