# Peer review of "ATOMS (Adjustable Transobturator Male System) Is an Effective and Safe Second-Line Treatment Option for Recurrent Urinary Incontinence after Implantation of an AdVance/AdVance XP Fixed Male Sling? A Multicenter Cohort Analysis"

_jcm, 2021, doi:10.3390/jcm11010081_

Round 1

Reviewer 1 Report

Various surgical concepts are available for treating stress urinary incontinence after radical prostatectomy. This study examines the use of ATOMS (adjustable transobturator male system) in a multicenter cohort analysis. ATOMS was applied after failure of an AdVance/AdVanceXP fixed male sling.

General comment:

The article is well written, clear and concise. The results support the conclusions.

Specific points:

The title could be confirmative, rather than formulated like a question, since the authors have clearly shown that ATOMS is ''an effective and safe second-line treatment option for recurrent urinary incontinence after implantation of an AdVance/AdVance XP sling''.

Since Table 1 is already mentioned in line 151, it would be more logic if it was placed before Figures 1 and 2, especially because Figures 1 and 2 are a visual representation of part of the data from the Table 1.

Advance XP is sometimes written separately, and sometimes jointly (AdvanceXP)

Line 193: the sentence should start with the word (''seven'') rather than a number since it is the beginning of the sentence.

Lines 236-238: the authors could briefly describe how the device that they used in their study differs from the first generation ATOMS Inguinal Port used in the reference 20.

Line 385: it is written ''mle'', instead of ''male''

Author Response

Please see document enclosed

Reviewer 2 Report

the study proposed by the authors is interesting and concerns an aspect of male incontinence still little investigated. in particular, the authors set themselves as objectives: the evaluation of effectiveness and safety of ATOMS as 117 secondary implant to treat male SUI after failed fMS Advance / Advance XP. there are some minor revisions that I would like the authors to give back: how was the bias managed carried out lying on the surgical technique? the operations were performed in different centers and each surgeon has his own experience. Moreover, why was a preliminary urodynamic examination not performed? 

Author Response

Please see document enclosed

Round 2

Reviewer 2 Report

It's ok.